# On the Importance of Expert Knowledge to Improve Foundation Models for Retinal Fundus Images

**Julio Silva-Rodríguez**[1]  JULIO-JOSE.SILVA-RODRIGUEZ@ETSMTL.CA

[1] *ÉTS Montreal*

**Hadi Chakor**[2]  HCHAKOR@DIAGNOS.CA

**Riadh Kobbi**[2]  RKOBBI@DIAGNOS.CA

[2] *DIAGNOS Inc.*

**Jose Dolz**[1]  JOSE.DOLZ@ETSMTL.CA

**Ismail Ben Ayed**[1]  ISMAIL.BENAYED@ETSMTL.CA

## Abstract

Foundation models are currently revolutionizing the medical image analysis community. Pre-trained on large data sources, such networks provide efficient transferability to downstream tasks. In this context, a myriad of foundation models leveraging large amounts of general medical data and increasing network sizes are appearing in the literature. In this short paper, we study the importance of incorporating domain-specific expert knowledge during pre-training of specialized foundation models in the context of fundus retina images. In particular, we focus on introducing the expert knowledge-driven vision-language model FLAIR (Silva-Rodriguez et al., 2023), comparing its benefits to larger-scale generalists and domain-specific self-supervised models. The pre-trained model is available at: https://github.com/jusiro/FLAIR .

**Keywords:** Foundation models, Fundus image, Vision-language pre-training.

## 1. Introduction

Vision-language models (VLMs), particularly CLIP (Radford et al., 2021), exploit large text-supervised data during pre-training. Nevertheless, natural image VLMs like CLIP may not capture fine-grained features and class hierarchies of medical images, which might be highly specialized concepts. Recently, medical VLMs have started to emerge. In particular, generalist models aim to assemble different medical image modalities (mostly radiology), to pre-train medical VLMs (Zhang et al., 2023). These datasets might contain limited data on specific modalities, such as fundus retina imaging, where text information is scarce and most datasets are categorically labeled. For this imaging modality, self-supervised pre-training has been the quick-fix solution to develop domain-specific foundation models (Azizi et al., 2023; Zhou et al., 2023). We argue that, even for categorically labeled images, VLMs are an appealing solution to integrate domain-specific expert knowledge, such as the dependencies between the categories, into visual representations.

## 2. Methods

FLAIR (Silva-Rodriguez et al., 2023) is a vision-language foundation model pre-trained at an assembly of 38 fundus open-access datasets with 288K samples and 101 conditions. $\mathcal{D}_T = \{(\mathbf{X}_n, y_n, \mathbf{T}_n)\}_{n=1}^{N}$ is composed of paired images, labels, and text descriptions, respectively.

**Vision-language-label pre-training.** The architecture is composed by a vision encoder, $\theta$, that projects images into $L^2$-normalized features, $\mathbf{u}$, and a text encoder, $\phi$, that analogously produces normalized embeddings, $\mathbf{v}$, from text descriptions. Pre-training consists of optimizing both encoders in mini-batches, $\mathcal{B}$, to align paired image and text descriptions with the same label category, following the next contrastive objectives:

$$\mathcal{L}_{i2t}(\theta, \phi, \tau | \mathcal{B}) = -\sum_{i \in \mathcal{X}_B} \frac{1}{|P_{\mathcal{T}_B}(i)|} \sum_{i' \in P_{\mathcal{T}_B}(i)} \log \frac{\exp(\mathbf{u}_i^T \mathbf{v}_{i'}/\tau)}{\sum_{j \in \mathcal{T}_B} \exp(\mathbf{u}_i^T \mathbf{v}_j/\tau)} \tag{1}$$

$$\mathcal{L}_{t2i}(\theta, \phi, \tau | \mathcal{B}) = -\sum_{j \in \mathcal{T}_B} \frac{1}{|P_{\mathcal{X}_B}(j)|} \sum_{j' \in P_{\mathcal{X}_B}(j)} \log \frac{\exp(\mathbf{u}_{j'}^T \mathbf{v}_j/\tau)}{\sum_{i \in \mathcal{X}_B} \exp(\mathbf{u}_i^T \mathbf{v}_j/\tau)} \tag{2}$$

where $\tau \in \mathbb{R}_{++}$ is a trainable scaling parameter, $|\cdot|$ denotes the cardinality of a set and $P_{\mathcal{T}_B}(i)$ and $P_{\mathcal{X}_B}(j)$ contain indices of similar-category subsets obtained by image labels.

**Expert knowledge.** Fundus datasets rarely contain text supervision. Therefore, we introduce a mapping function, which generates *domain expert knowledge* descriptions from the categorical labels based on clinical ophthalmology literature. This transformation maps a given category, $y^*$, to an ensemble of descriptions of relevant findings or inter-category relationships such that $\{\mathbf{T}^*\}_1^P = \pi_{EK}(y^*)$. For example, a text description of category "*proliferative DR*" would be "*contains neovascularization*", while the category "*exudates*" could be described as "*small white or yellowish-white deposits with sharp margins*".

## 3. Experiments

**Datasets.** A wide range of color fundus analysis tasks is addressed: diabetic retinopathy grading using MESSIDOR (Decencière et al., 2014) and DeepDRID (Liu et al., 2022), multiple diseases in FIVES (Jin et al., 2022), glaucoma detection in REFUGE (Orlando et al., 2019), myopic maculopathy grading in MMAC (Li et al., 2024), and bi-disease differentiation in FLAIR's partitions 20x3 (Cen et al., 2021) and ODIR$_{200x3}$. The evaluation is carried out using a 5-fold cross-validation with 20% of testing data and balanced average accuracy.

**Baselines.** We employ recently released foundation models. We include CLIP models (Radford et al., 2021), vision-language models pre-trained on large natural image sources. Also, we include a medical generalist model, *i.e.* BioMedCLIP (Zhang et al., 2023), pre-trained on 15M heterogeneous medical image and text pairs. Also, RETFound (Zhou et al., 2023), a domain-specific, self-supervised model for fundus retina images, is evaluated. This model is pre-trained on 800K images via Masked Autoencoder loss.

**Transferability.** First, vision-language models are evaluated at **Zero-Shot** (ZS) classification, using an assembly of text prompts per class. Second, the transferability of the pre-trained visual representations is assessed by **Linear Probing** (LP), using the same multi-class logistic regression optimizer as in CLIP, *i.e.*, L-BFGS (Nocedal, 1980). Finally, we evaluate the effect of fully **Fine-Tuning** (FT) the pre-trained model on the target task.

## 4. Results and discussion

**Results.** Table 1(a) shows that, albeit BiomedCLIP outperforms **Zero-Shot** CLIP, it does not provide meaningful predictions on domain-specific fine-grained tasks. In contrast, FLAIR largely outperforms such methods across all tasks. Regarding **Linear Probing**, Table 1(b) delves into the limitations of generalist medical models such as BiomedCLIP, which shows worse transferability than natural image pre-trained models. Interestingly, this is also the case of the recently popularized RETFound. In contrast, FLAIR can be efficiently adapted with a lightweight LP across all tasks, even if target diseases have not appeared used during pre-training, *e.g.*, ODIR$_{200x3}$, or MMAC. Domain-specific self-supervised models such as RETFound largely rely on **Fine-Tuning** during adaptation. We show in Figure 1 that this strategy might provide good results on in-distribution data, but potentially deteriorate the generalization performance on OOD distributions (Kumar et al., 2022).

Table 1: **Transferability results.**

| (a) *Zero-shot* | | MESSIDOR | FIVES | REFUGE | 20x3 | ODIR$_{200x3}$ | MMAC | **Avg.** |
|---|---|---|---|---|---|---|---|---|
| CLIP | ViT-B/32 | 0.200 | 0.256 | 0.433 | 0.333 | 0.480 | 0.183 | 0.314 |
| BiomedCLIP | ViT-B/16 | 0.207 | 0.415 | 0.624 | 0.617 | 0.583 | 0.274 | 0.453 |
| FLAIR | RN50 | **0.604** | **0.735** | **0.883** | **0.983** | **0.667** | **0.400** | **0.712** |
| (b) *Linear Probing* | | | | | | | | |
| ImageNet | RN50 | 0.424 | 0.741 | 0.733 | 0.983 | 0.887 | 0.631 | 0.733 |
| CLIP | ViT-B/32 | 0.491 | 0.800 | 0.720 | 0.950 | 0.917 | 0.642 | 0.753 |
| BiomedCLIP | ViT-B/16 | 0.433 | 0.654 | 0.776 | 0.866 | 0.883 | 0.678 | 0.715 |
| RETFound | ViT-B/16 | 0.457 | 0.765 | 0.747 | 0.950 | 0.887 | 0.547 | 0.725 |
| FLAIR | RN50 | **0.719** | **0.879** | **0.843** | **1.000** | **0.935** | **0.740** | **0.852** |

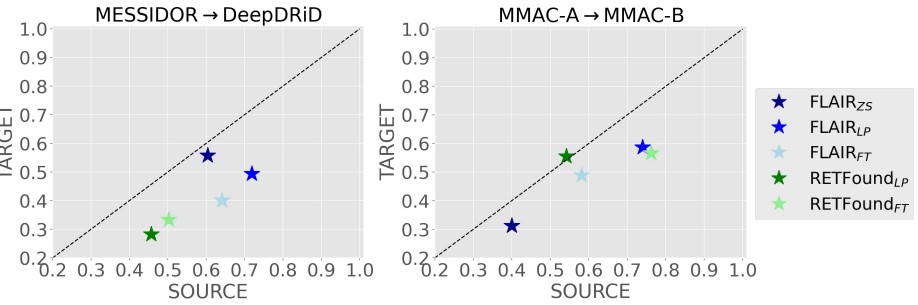

Figure 1: **Fine-Tuning and Domain Generalization.**

**Discussion.** Recently introduced foundation models based on large generalist medical sources or unsupervised domain-specific pre-training fail to provide efficient transferability on fine-grained fundus retinal diagnosis tasks. If such models require full Fine-Tuning, they lose the underlying benefit of the foundation models: the data- and resource-efficient adaptation to challenging clinical contexts. Thus, introducing available open-access domain-specific knowledge via labels and text descriptions provides a more appealing direction.

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
