# OpenReview forum: "On the Importance of Expert Knowledge to Improve Foundation Models for Retinal Fundus Images"
_MIDL.io/2024/Short_Papers — MIDL 2024 Short Papers_

### Official Review · Reviewer_3HkY · 2024-04-22

**Confidence:** 4
**Final Rating:** 5

**Review:**

Summary
-------

This paper is a short version of an existing preprint. The authors developed a language vision model for fundus images, trained on 37 publicly available retinal fundus datasets.

Strengths
---------

- The paper is well written and a useful summary of the preprint it is based on. The preprint itself seems of high quality and contains a comprehensive description of the datasets used, the model development, and its experimental evaluation and comparison against other generalist and specialised foundation models.
- The proposed model seems to perform well and I would expect it to have high clinical impact.

Weaknesses
----------

- I don't see any weaknesses, this is a solid submission and I believe well suited for the short-paper track.

---

### Decision · Program_Chairs · 2024-04-26

Accept